# Phytochemical Characterization of a Tree Tomato (*Solanum betaceum* Cav.) Breeding Population Grown in the Inter-Andean Valley of Ecuador

**DOI:** 10.3390/plants11030268

**Published:** 2022-01-20

**Authors:** William Viera, Iván Samaniego, Diana Camacho, Nasratullah Habibi, Lenin Ron, Naveedullah Sediqui, Javier Álvarez, Pablo Viteri, Andrea Sotomayor, Jorge Merino, Wilson Vásquez-Castillo, Beatriz Brito

**Affiliations:** 1National Institute of Agricultural Research (INIAP), Santa Catalina Research Site, Fruit Program, Tumbaco Experimental Farm, Tumbaco 170902, Ecuador; william.viera@iniap.gob.ec (W.V.); pablo.viteri@iniap.gob.ec (P.V.); andrea.sotomayor@iniap.gob.ec (A.S.); jorge.merino@iniap.gob.ec (J.M.); 2Department of Nutrition and Quality, Santa Catalina Research Site, National Institute of Agricultural Research (INIAP), Cutuglahua 171107, Ecuador; ivan.samaniego@iniap.gob.ec (I.S.); javier_alvarezm84@hotmail.com (J.Á.); 3Faculty of Chemistry Sciences, Universidad Central del Ecuador (UCE), Quito 170521, Ecuador; dkcd1994@gmail.com; 4Faculty of Agriculture, Balkh University, Balkh 1702, Afghanistan; nasratullah.habibi14@gmail.com; 5Faculty of Veterinary Medicine and Zoothecnics, Universidad Central del Ecuador (UCE), Quito 170521, Ecuador; ljron@uce.ude.ec; 6Faculty of Agriculture, Alberoni University, Kohistan 1254, Afghanistan; naweedone@gmail.com; 7Agroindustry and Food Science, Universidad de las Américas (UDLA), Quito 170503, Ecuador; 8Independent consultant, Quito 170103, Ecuador; bdbg61@gmail.com

**Keywords:** antioxidants, pulp, polyphenols, flavonoids, carotenoids, anthocyanins, antioxidant activity

## Abstract

Tree tomato (*Solanum betaceum* Cav.) is an Andean fruit crop that is grown in Ecuador. It is an exceptional source of minerals and vitamins, thus has nutraceutical properties. The objective of this research was to carry out a phytochemical characterization of a breeding population composed of 90 segregants. Pulp (including mesocarp, mucilage, seeds and placenta) was ground and sieved in order to obtain the liquid pulp to be lyophilized for the chemical analyzes. Antioxidants compounds were determined by spectrophotometry and vitamin C by reflectometry. Data were analyzed by principal components, grouping, and variance analyses; in addition, Z Score estimation was carried out to select elite individuals. There was a broad variability in the data obtained for the breeding population, polyphenol content varied from 5.11 to 16.59 mg GAE g^−1^, flavonoids from 1.24 to 6.70 mg cat g^−1^, carotenoids from 50.39 to 460.72 µg β-carotene g^−1^, anthocyanins from 1.06 to 240.49 mg cy-3-glu 100 g^−1^, antioxidant capacity from 49.51 to 312.30 µm Trolox g^−1^, and vitamin C from 78.29 to 420.16 mg 100 g^−1^. It can be concluded that tree tomato is a good source of beneficial biocompounds and has a high antioxidant capacity.

## 1. Introduction

Despite the existence of a large number of fruits around the world, there are a high number of underexplored native and exotic fruit species.This can become an opportunity to access special markets where consumers have an interest in the presence of functional compounds potentially capable of preventing diseases [1,2]; one of these fruits is the tree tomato (*Solanum betaceum* Cav.).

Tree tomato grows in Ecuador at altitudes from 2000 to 2800 masl in the highlands and from 100 to 1500 masl in the Amazon region [3]. It is a fruit that contains polyphenols, anthocyanins, carotenoids, vitamins A, B6, C, and E, and is rich in iron, potassium, fiber, and other important minerals for human health [4]. Studies show that regular fruit consumption can prevent various diseases and disorders due to the presence of bioactive compounds with antioxidant properties [5,6,7]. The role of antioxidants has gained increasing interest due to their beneficial health effects associated with several chronic diseases, including cardiovascular and bone diseases, and certain types of cancer [8,9]. This benefit would be associated with their property of giving greater cellular protection against oxidation [10]. Polyphenols can neutralize reactive oxygen species [11], flavonoids have the antioxidant, antibacterial and antiviral effect [12], carotenoids have nutraceutical properties and also are the responsible for fruit color [13], and anthocyanins can prevent lipid oxidation [14].

Various studies refer to the beneficial effects of biocompounds such as polyphenols, flavonoids, carotenoids, and anthocyanins obtained from fruits, due to their anticancer, cardioprotective, antidiabetic, neuroprotective [15,16], lipid antiperoxidative [17], antiallergic, antiatherogenic, anti-inflammatory, antimicrobial, antithrombotic, vasodilator effects [11,12], and can delay the damage produced at the central nervous system as a consequence of cell aging [18].

Tree tomato has been reported to be a good source of these functional compounds [19,20,21,22,23] together with vitamin C (VC), having a high antioxidant capacity [24]. Some studies about phytochemical characterization of local cultivars have been carried out in Ecuador [19,21,24] but this kind of study has not been conducted in breeding populations. The aforementioned research has proven that the tree tomato is a good source of antioxidant compounds. However, it has been reported that the concentration of these biocompounds varies depending on environmental factors, genetics, and stage of fruit maturity [25,26].

The Fruit Program of the National Institute of Agricultural Research (INIAP) (Quito, Ecuador) has been working on this fruit crop, evaluating offspring that comes from interspecific crosses of *S. betaceum* X *S. unilobum* with backcrossing to *S. betaceum*, generating progeny of high phenotypic variability [27,28], but its fruit has not been phytochemically characterized. Therefore, the objective of this research was to characterize the compounds and antioxidant capacity of a population of tree tomato segregants to identify elite individuals for future breeding programs.

## 2. Results and Discussion

### 2.1. Physical and Chemical Traits

Physical and chemical properties of the tree tomato varies with the cultivars and the geographical origin [23,29]. For example, research has shown that Ecuadorian yellow orange and purple red tree tomato cultivars were larger and softer than the same cultivars grown in Spain; however, the chemical composition was similar [30].

Phenolic compounds have been found in greater amount in purple red than golden-yellow cultivars [30] and their content is higher in the pulp than in the peel of tree tomato [31]. The accumulation of phenols, carotenoids, vitamin C, and other bioactive compounds are influenced by environmental factors [22]. Low temperature promotes the accumulation of these compounds in some fruit during the growth stage [32]. Therefore, this factor could influence the data obtained in the tree tomato population because the temperature generally decreases from 25 to 10 °C in the Tumbaco Experimental Farm of INIAP.

Table 1 and Figure 1 show the variation observed in the different traits recorded in the 90 segregants of tree tomato, which indicates that there was a wide variability in this study (Appendix A).

Studies have found that total polyphenols (TP) are higher in the peel than in the pulp of tree tomato [30,33]. In New Zealand’s varieties (red, purple red, and yellow pulp), TP ranged between 8.75 and 7.07 mg GAE g^−1^, values that are included in the range (5.11 and 16.59 mg GAE g^−1^) found in the evaluated breeding population; these compounds were the most related to the antioxidant capacity in this fruit crop [22]. The TP values found in this study were higher than those (1.17 to 1.91 mg GAE g^−1^) found by [34] in New Zealand cultivars (yellow and red); and also higher than those (2.4 to 6.2 mg GAE 100 g^−1^) reported by [21] in Ecuadorian cultivars (orange, red, and purple cultivars). The average population value (8.47 mg GAE g^−1^) is higher than 1.01 mg GAE g^−1^ reported by [35] in the yellow cultivar grown in Peru.

Flavonoid compounds are considered to be the largest group of naturally occurring phenols [20] and they are used for the prevention and cure of diseases associated with free radicals [36]. It has been found that tree tomato pulp contains a greater number of flavonoids than its peel [31]. The average population value (2.99 mg cat g^−1^) was found to be lower than reported by [20] with 6.44 mg cat g^−1^ in a Malaysian cultivar. On the other hand, quercetin and myricetin (bioflavonoids) have only been detected in the peel of tree tomato [30].

Tree tomato is rich in carotenoids [21] and β-carotene has been reported as the dominant carotenoid in this fruit crop [37]; however, other carotenoids such as β-cryptoxanthin, lutein and chlorophyll pigments have also been found [38]. The same author found that carotenoid content was higher in the pulp than in the peel. A total carotenoid (TC) range from 242.90 to 650.00 µg β-carotene g^−1^ by [30] and from 36.8 to 40.8 µg β-carotene g^−1^ by [37] have previously been reported; the former was higher than this study (50.39 to 460.72 µg β-carotene g^−1^) but the latter was lower, both were in Ecuadorian cultivars. The average value (223.38 µg β-carotene g^−1^) was higher than the content reported by [4] in a Malaysian cultivar and by [39] in a Brazilian cultivar, with 48.00 and 102.50 µg β-carotene g^−1^, respectively. The highest population value (460.72 µg β-carotene g^−1^) was much higher than that found by [35] in the yellow cultivar grown in Peru (42.70 µg β-carotene g^−1^).

Anthocyanins in tree tomato could act as a natural food additive to extend shelf life by preventing or delaying lipid oxidation and also to improve food quality and its nutritive value [40,41]. Total anthocyanin content (TAC) is high in the red and purple tree tomato types [19]; however, it has been reported that the tree tomato pulp contains a small number of anthocyanins and they are mainly in the peel [33], even if [31] found the opposite. In this study, a TAC from 0 (no detected value) to 240.49 mg cy-3 glu 100 g^−1^ was found, which is higher than that reported by [34] with 0 (yellow cultivar) to 82.00 (red cultivar) mg cy-3 glu 100 g^−1^.

This fruit crop has a greater antioxidant capacity than many antioxidant-rich fruits, indicating that its phenolic compounds are stronger antioxidants [24]. The same authors [24] analyzed different cultivars grown in Ecuador and found an antioxidant capacity from 10 to 50 µmol Trolox g^−1^ (FRAP) and from 22 to 89 µmol Trolox g^−1^, values that are lower than those found in this study with 52 to 361 µmol Trolox g^−1^ (FRAP) and from 49 to 312 µmol Trolox g^−1^ (ABTS).

This fruit is a good source of vitamin C [24]. A range from 78.29 to 428.19 mg 100 g^−1^ in the analyzed population was found; these values were higher than the range found by [42] (29.8 to 31.0 mg 100 g^−1^) and the value reported by [34] (24.7 mg 100 g^−1^) in yellow and red New Zealand cultivars. They were also higher than concentrations reported by [30] (17 and 16 mg 100 g^−1^) in golden-yellow and purple-red tree tomatoes from Ecuador and by [4] in reddish-brown skin tree tomato from Malaysia (55.9 mg 100 g^−1^).

It was stated that total soluble solids (SS) and titratable acidity (TA) are similar between golden yellow and purple-red tree tomato varieties [30]. There was a range of 7.00 to 12.9 °Brix and for titratable acidity of 0.68% to 1.63% in the population evaluated; values which are similar to that found in literature [29].

Pulp brightness is a parameter that remains stable once the fruit is harvested and then stored [43]. This parameter showed a variation of 56.64% in the whole population.

According to the confidence intervals estimated for each variable using all data from the 90 segregants, it was possible to set three categories (Table 2). This categorization can be used as a reference to compare further breeding populations or cultivars currently grown in different sites of Ecuador or overseas; however, it has been considered that biocompounds content is also related to environmental and genetic factors [25,26].

### 2.2. Principal Component Analysis (PCA)

In the PCA analysis, the first two components explained 65.68% of the variance observed in the data (Figure 2) and the third component explained 16.39% more, thus these three components explained the 82.07% of the variance observed in the data. The first accounting of 40.74% of explained variance was a contrast between carotenoids (0.81) and soluble solids (0.89) vs. antioxidant capacity (ABTS −0.96 and FRAP −0.97), which means that the higher the antioxidant capacity, the lower the content of carotenoids and soluble solids in the pulp. Polyphenol content was the only compound that had a positive and significant (*p* < 0.001) partial correlation value (0.53) with ABTS, which means that this is the main phenolic compound influencing antioxidant activity; however, phenolic compound content is affected by temperature and by soluble solids [44]. On the other hand, these results would indicate that carotenoids are not influencing antioxidant capacity, they obtained a low and negative partial correlation value (Appendix A).

### 2.3. Factor Analysis (FA)

FA (Table 3) identified three main factors. The first factor was called “antioxidant capacity”, which was strongly influenced by ABTS and FRAP, and negatively affected by carotenoid and soluble solids content, thereby confirming the PCA results. This factor supports that tree tomato has an appreciable antioxidant capacity [24], but the main factor that influenced antioxidant capacity was the polyphenol content as was previously mentioned.

The second factor was called “appearance”, which was influenced by brightness and b (fruit color parameters) while anthocyanin content negatively affected it. Anthocyanins are the bright colored pigments found in fruits [45]; however, they were in low amount in a great proportion of the segregants evaluated in this study (Figure 3). Finally, the third factor was called “maturity”, which was influenced by the sugar/acid ratio and oppositely by titratable acidity, which is a parameter that decreases over the post-harvest time, while the opposite occurs with soluble solids [46]; however, both are included in the estimation of the sugar/acid ratio.

### 2.4. Z Score Estimation

The tree tomato has therapeutic and nutritional values that can satisfy the demands of health-conscious consumers; however, it is an underutilized species [29]. This fruit can be used in pharmaceutical applications, constituting a promising commercial crop with the potential to be fully exploited as a source of healthy products [23,29]. In Ecuador, few breeding studies have been carried out in tree tomatoes [27,28,47,48,49,50]; for this reason, selecting individuals showing high biocompounds contents are useful to set further breeding programs and in developing new varieties [21].

Table 4 shows the 22 segregants that obtained a Z Score index ≥ 2.00 in at least one of the variables evaluated. Some individuals stood out, which obtained this kind of value in more than two traits, such as the segregant 47 that showed values for TP (2.60), TF (2.99), and TAC (3.91) and 84 that obtained outstanding values for TP (4.27), TF (2.99), ABTS (2.16), and FRAP (2.84).

Functional compounds have nutraceutical properties and these effects could be due to compounds such as polyphenols and their combinations with other bioactive compounds in the fruit [29]. For this reason, elite individuals were identified by combining TP, TC, and VC by the Z Score; individuals with an index > 1 were selected (Table 5).

It was observed that the 10 selected segregants were also part of the 23 individuals that showed outstanding values in various parameters (shown in Table 4). In particular, segregant 84 stood out because it had a Z Score index higher than 2 (*p* < 0.01), which means that this individual is superior to the others, mainly in TP (Table 4) that was the parameter with the highest weighting for the combined Z Score. This segregant would be less acid according to the TI value (1.20), index that has been used in other fruits [51] and in this study showed a high correlation with TA (0.99). However, segregant 84 has yellow pulp which is not currently accepted by Ecuadorian consumers, preferring orange cultivars [27], but would have export potential to other countries. An antioxidant capacity of 4.65 µmol trolox g^−1^ (ABTS) and 28.83 mg 100 g^−1^ of vitamin C cultivated in Peru (yellow cultivar) [35] has been reported, which is much lower than the value obtained by the selected segregant 84 (312.30 µmol trolox g^−1^ and 251.66 mg 100 g^−1^).

Segregant 4 which has a slightly acid taste (TI = 1.73) and orange color is a good option to be used as a parental for further breeding (crosses with the local cultivar Giant Orange) due to its good content of TP (9.65 mg GAE g^−1^), TC (454.95 µg β-carotene g^−1^) and VC (289.54 mg 100 g^−1^). On the other hand, individuals with purple pulp can be used to generate breeding populations to develop new varieties with export potential because of their good content of antioxidant compounds and overseas acceptance.

### 2.5. Variance Analysis (ANOVA)

After grouping based on antioxidant compounds and pulp color, eight groups were obtained in both cases. ANOVA was carried out independently for each kind of grouping.

Table 6 shows ANOVA for the groups according to the antioxidant compound criteria. It was observed that group 2 was statistically different (*p* > 0.05), showing the highest value of TP (12.64 mg GAE g^−1^), TF (4.76 mg cat g^−1^), and TAC (233.79 mg cya-3-gly 100 g^−1^). In terms of TC, groups 3 and 4 were similar, showing the highest values with 402.39 and 389.35 µg β-carotene g^−1^, respectively. The group 3 showed the highest value of VC (358.49 mg 100 g^−1^) and TI (1.90). Turning to the antioxidant capacity, group 1 showed the highest value of FRAP (292.81 µm Trolox g^−1^) and ABTS (224.77 µm Trolox g^−1^).

Fruit color has proven to be a highly heritable trait because it is mainly controlled by a few dominant genes [52]. Table 7 shows the ANOVA for the groups according to pulp color criteria. It was observed that group 5 was statistically different (*p* > 0.05), showing the highest value of TP (12.64 mg GAE g^−1^), TF (4.76 mg cat g^−1^), and TAC (233.79 mg cy-3-glu 100 g^−1^). In terms of TC, group 8 obtained the highest value (339.66 µg β-carotene g^−1^). Groups 6 and 8 showed the highest values of VC (311.57 and 292.49 mg 100 g^−1^, respectively). Finally, for the antioxidant capacity, group 4 showed the highest value of FRAP (295.59 µm Trolox g^−1^) and ABTS (206.83 µm Trolox g^−1^).

Group 2 obtained the highest value for TP, TF, and TAC according to the antioxidant compound criteria, whereas group 5 was according to the pulp color criteria. Nevertheless, segregants 51, 66, 74, 78, and 90 (all orange yellow pulp color) were present in both groups; consequently, they can be considered as selected individuals taking into consideration both criteria. In terms of antioxidant capacity, group 1 was the best according to the antioxidant criteria, while group 4 was in the pulp color criteria. All segregants conforming to group 4 had orange pulp; thus, they can be considered as promising individuals for this trait, which is the Ecuadorian preference.

## 3. Materials and Methods

### 3.1. Plant Material

The original breeding population had 267 segregants (plants grafted in *Nicotiana glauca*) in 2016 [27], 158 plants persisted in 2018 [28], and finally in 2019, 90 segregants remained alive (because of the incidence of diseases, mainly anthracnose) and showed phenotypic variability. Consequently, samples of these 90 tree tomato segregants, which showed a diversity of pulp colors (ranging from yellow, orange, and purple), were analyzed. The fruit was harvested in the state of edible maturity. The loss of green pigmentation (100% change in peel color) and the peduncle showing a woody dark brown color were taken into consideration, according to the scale set in the NTC 4105 standard [53]. The field sample consisted of 300 g of fruit from each tree tomato segregant. In the laboratory, fruits were washed, peeled and the pulp (including mesocarp, mucilage, seeds and placenta) was ground and sieved in order to obtain just the liquid pulp to be lyophilized for the chemical analyses.

### 3.2. Chemical Reagents

Deionized water obtained through a Milli-Q Academic water purification system (Millipore, Sao Paulo, Brazil) was used. The standards used in the study were obtained from Sigma-Aldrich (St. Louis, MO, USA), and analytical grade solvents and reagents were obtained from Merck (Darmstadt, Germany), and were: (+) catechin, gallic acid, cyanidin-3-glucoside chloride, ABTS (2,2-azinobis-3-ethyl-benzothiazoline-6-sulfonic acid), and Trolox (6-hydroxy-2,5,7, 8-tetramethylchroman-2-carboxylic acid).

### 3.3. Preparation of Samples

The fruits were dried by lyophilization (Figure 3), ground in a Retsch model ZM 200 mill (Hann, Germany), and passed through a stainless-steel sieve (1 mm mesh) to ensure a uniform particle size.

### 3.4. Color Evaluation

Pulp color was registered by a ColorTec-PCM handheld colorimeter (ColorTec, Clinton, NJ, USA) with a measurement angle of 10°, Illuminator D65, and aperture of 8 mm. The chromatic properties were determined using the *L* * *a* * *b* * color method of the CIE (Commission Internationale de l’Eclairage) and were expressed in terms of *L* (lightness), *a* (red/green), and *b* (blue/yellow) coordinates. An amount equivalent to 200 g of pulp was homogenized in a blender, then 30 g was placed in a Petri dish, avoiding the formation of lumps and bubbles. The Petri dish was placed on a white surface and divided into four equal parts. Measurements were performed in triplicate in each quarter and at the center of the plate.

Hue angle (°*H*) was calculated by the formula [54]:(1)$${}^{{}^{\circ}}H=\tan^{-1}\left(\frac{b}{a}\right)$$
where °*H* is the Hue angle, tan is tangent function, *a* is the value of parameter *a*, and *b* is the value of parameter *b*.

Chroma (*C*) was estimated based on the following formula [54]:(2)$$C={\left(a^{2}+b^{2}\right)}^{1/2}\;$$
where *C* is Chroma, *a* is the value of parameter *a*, and *b* is the value of parameter *b*.

### 3.5. Determination of Vitamin C (VC), Soluble Solids (SS), Titratable Acidity (TA), Sugar/Acid Ratio (SAR), and Taste Index (TI)

VC (L-ascorbic acid) was measured with a reflectometer RQflex plus 10 (Merck, Germany) that uses the principle of reflectance photometry, in which the intensity of the light reflected by test strips (Reflectoquant™, Merck, Kenilworth, NJ, USA) is measured and related to the concentration of ascorbic acid. Thus, 30 g of pulp was mixed with 200 mL of water, a test strip was introduced in the solution for 2 sec and then it was placed in the reflectometer for 15 s and the content of ascorbic acid was recorded in mg L^−1^. The content of VC was expressed as mg of ascorbic acid 100 g^−1^ fresh weight (FW), using the formula [54]:(3)$$\mathrm{VC}=\frac{L\times V}{W}$$
where VC is the content of vitamin C, *L* is the reflectometer lecture (mg L^−1^), *V* is the final volume and *W* is the sample weight.

SS concentration was determined by refractometry using an ATAGO digital refractometer (Tokyo, Japan). Two drops of fruit juice were placed on the prism of the equipment surface, and the percentage of soluble solids was shown directly, expressed in terms of °Brix. TA was measured by potentiometric titration. A total of 30 g of fruit pulp was weighed and brought to a volume of 200 mL with distilled water. Subsequently, 20 mL was placed in a 25 mL beaker and titrated with a 0.1 N NaOH solution until pH 8.2 was reached. The results were reported based on citric acid content. SAR was determined by the formula:(4)SAR=SS/TA
where SAR is the sugar/acid ratio, SS is soluble solids (°Brix) and TA is titratable acidity (g citric acid 100 g^−1^).

Finally, TI was determined by the following formula [55]:(5)$$\mathrm{TI}=\left(\mathrm{SS}/20\;\mathrm{TA}\right)+\mathrm{TA}$$

### 3.6. Extract Preparation

Phytocomponents extraction was carried out following the method by [56]. A sample weighing 0.3 g of dry weight was placed in 15 mL plastic centrifuge tubes, and 5 mL of a methanol/water/formic acid solution (70:30:0.1 *v*/*v*/*v*) was added. The sample was subject to a shaking extraction process in a vortex (Mistral Multi-Mixer, Melrose Park, IL, USA) for 5 min and subsequently placed in an ultrasound bath (Cole-Palmer model 8892, Chicago, IL, USA) at room temperature (20 °C) for 10 min with a frequency of 47 kHz and a power of 200 W. The sample was centrifuged in a Damon IEC/Division centrifuge (Needham Hts., Needham, MA, USA) for 10 min at 5500 rpm. The supernatant was transferred to a 25 mL amber volumetric balloon. This process was repeated four more times and brought to volume with the extraction solution. The same extract was used for the antioxidant activity (AA) determination.

### 3.7. Quantification of Total Polyphenols (TP)

TP quantification was performed by UV-visible spectrophotometry according to [57]. A volume of 1 mL of diluted extract was placed in a 15 mL test tube; 6 mL of distilled water and 1 mL of Folin–Ciocalteu reagent were added, and the mixture was left to rest for 3 min. Subsequently, 2 mL of 20% Na_2_CO_3_ (*w*/*v*) was added and heated at 40 °C for 2 min. This reaction formed a blue chromophore; its absorbance was measured at 760 nm on a Shimadzu UV-VIS model 2600 spectrophotometer (Shimadzu, Kyoto, Japan). Five extraction cycles were needed to get 100% TP recovery. Results were expressed as milligrams of gallic acid equivalents (GAE) per gram of dry weight (mg GAE g^−1^ DW).

### 3.8. Quantification of Total Flavonoids (TF)

TF was quantified by UV-visible spectrophotometry, according to [57]. A volume of 1 mL of the diluted extract was placed in a 15 mL tube; 4 mL of distilled water was added, and the mixture was homogenized. Then, 0.3 mL of 5% sodium nitrite (*w*/*v*) and 0.3 mL of 10% aluminum chloride (*w*/*v*) were added, and allowed the sample to stand for 5 min after the addition of each reagent. Finally, 2 mL of 1N NaOH was added, and the volume was made up to 10 mL with distilled water. This reaction formed a pink chromophore, its absorbance was measured at 490 nm with a Shimadzu UV-VIS model 2600 spectrophotometer (Shimadzu, Kyoto, Japan). Five extraction cycles were needed to reach 100% TF recovery. Results are expressed as milligrams of catechin equivalents per gram of dry sample (mg cat g^−1^ DW).

### 3.9. Quantification of Total Carotenoids (TC)

TC determination was performed following the method described by [58]. One gram of lyophilized sample was mixed with 1 g of calcium chloride and a solution of hexane, acetone, ethanol, and BHT in proportion 50:25:25:0.1 (*v*:*v*:*v*:*v*) to carry out the extraction. The mixture was stirred in a vortex (Mistral 4600, Multi-Mixers; Melrose, IL, USA), and centrifuged. The extract was transferred to a separatory funnel and the hexane phase was recovered in a 25 mL volumetric balloon. Finally, the extract was passed to a glass cell to read the absorbance at 490 nm on a Shimadzu UV-VIS model 2600 spectrophotometer (Kyoto, Japan). The results were expressed as micrograms of β-carotene for each gram of dry sample (DW).

### 3.10. Quantification of Total Anthocyanin Content (TAC)

TAC was calculated by UV-visible spectrophotometry, following the pH differential methodology proposed by [59]. An amount weighing 0.3 g of lyophilized sample was placed in a 15mL centrifuge tube; 5 mL of pH 1.0 buffer (KCl 0.2 N and HCl 0.2 N) was added and stirred in a Mistral Multi-Mixer vortex (Melrose Park, IL, USA) for 5 min then in a Cole Palmer model 8892 ultrasound bath (Chicago, IL, USA) and centrifuged at 5500 rpm for 10 min. The supernatant was transferred to a 25 mL amber volumetric balloon. This procedure was repeated three more times, and the mixture was brought to volume with the pH 1.0 buffer solution. Extraction with pH 4.5 buffer (CH_3_COONa 1M and HCl 1N) was performed using the same methodology. Quantification was performed by measuring the absorbance in the extracts (pH 1.0 and 4.5) at two wavelengths (510 and 700 nm) by a Shimadzu UV-VIS model 2600 spectrophotometer (Shimadzu, Kyoto, Japan). Seven extraction cycles were needed to reach 100% TAC recovery. Results were expressed as milligrams of cyanidin-3-glucoside chloride per gram of dry weight (mg cy-3-glu g^−1^ DW).

### 3.11. Antioxidant Activity (AA)

AA was evaluated by the 2,2-azinobis (3-ethyl-benzothiazoline-6-sulfonic acid) cation bleaching method (ABTS+) according to [57]. The ABTS+ solution (7 mM) and the potassium persulfate solution (2.45 mM) were mixed in a 1:1 ratio (*v*/*v*). The next day, the absorbance of the previously prepared ABTS+ working solution was measured, and it was diluted with phosphate buffer until obtaining an absorbance of 1.1+/−0.01 at 734 nm. Volumes of 200 µL of the samples were placed in 15 mL test tubes, and 3.8 mL of the ABTS+ working solution was added. The solution was left to stand for 45 min, and the final absorbance at 734 nm was measured using a Shimadzu UV-VIS model 2600 spectrophotometer (Shimadzu, Kyoto, Japan). Results are reported as micromoles of equivalent Trolox per gram of dry weight (µmol Trolox g^−1^ of sample DW).

In addition, AA was also measured by the ferric reducing power (FRAP) method following the methodology described by [57]. A volume of 1.0 mL of diluted extract was placed in a 15 mL test tube; 2.5 mL of phosphate buffer at pH 6.6 and 2.5 mL of a 1.0% potassium ferrocyanide solution were added. The mixture was shaken and incubated at 50 °C for 20 min. Subsequently, 2.5 mL of 10% trichloroacetic acid was added along with 2.5 mL of water and 0.5 mL of 1% FeCl3. The mixture was homogenized in a vortex (Mistral Multi-Mixer, Melrose Park, IL, USA). It was left to stand for 30 min in the dark, and a green complex (ferrous chloride–potassium ferrocyanide) was formed, and the absorbance at 700 nm was measured in a Shimadzu UV-VIS model 2600 spectrophotometer (Shimadzu, Kyoto, Japan). Results are expressed as micromoles of equivalent Trolox per gram of dry weight (DW) (µmol Trolox g^−1^ of sample DW). 

### 3.12. Statistical Analysis

Data were analyzed using the statistical software R 4.1.1 [60]. Principal component analysis was carried out to visualize the relationship among traits (antioxidant traits plus soluble solids and titratable acidity), taking into account the correlation between variables. A Factor analysis was carried out to determine which variables mainly affect the principal fruit characteristics. To carry out the analysis of variance, individuals were grouped based on two criteria (antioxidant traits and color parameters); the k-means method, which takes into consideration the Elbow method to define the number of groups [61]. Polyphenols, flavonoids, carotenoids, anthocyanins, FRAP, ABTS, and vitamin C were used for grouping by the antioxidant criteria, whereas the color parameters *L*, *a* and *b* were used for the color criteria. Tukey test (5%) was used to determine differences between means.

Z Scores were estimated to select individuals showing a value ≥ 2 (*p* < 0.01) for independent traits. However, according to the literature, polyphenols and carotenoids have a considerable relevance because of their beneficial for human health [62,63], while vitamin C is an essential antioxidant molecule in plants and fruits are the main source of it [64]. These three traits were selected to calculate a combined Z Score because they were considered as relevant antioxidants compounds, and they were assigned a weight as follows: the content of polyphenols (50%), carotenoids (30%) and vitamin C (20%); in this case, segregants showing a value > 1 (*p* < 0.05) were selected as élite individuals.

The total Z Score [65] was composed based on the participation of the individual Z Scores as follows:(6)$$Z=(\mathrm{Xi}\;-\bar{X}{)/\mathrm{SD}}_{X}$$
where X*i* is the sample value, $\bar{X}$ is the sample mean, and SDx is the sample standard deviation.

The Z Score index is given by the following expression:(7)$$Z_{j}=\underset{i}{\sum }w_{i}Z_{i}$$
where *Z_i_* is the Z Score of each of the variables in participation (standardization of the variables) and *w_i_* is the weight assigned to each variable. Finally, confidence intervals for each of the variables were estimated to set a categorization (low, medium, and high) of the contents of each compound. The mean ± margin of error was used to calculate the two confidence intervals, according to the following formula:
(8)$$\mathrm{CI}=\;\bar{X}\;\pm Z_{c}\left(\frac{s}{\sqrt{n}}\right)$$
where CI is a confidence interval, $\bar{X}$ is the sample mean, *Z_c_* is the value for confidential level, *s* is the sample standard deviation and *n* is the number of elements in a simple. The values between minimum data value and the first confidence interval were used to set the “low” category; values from the first interval until second interval (including mean) were used for the “medium” category, while values of the second confidence interval until maximum data value were used for the “high” category.

## 4. Conclusions

Tree tomato fruit is a good source of biocompounds and has high antioxidant capacity, consequently, it can be considered as a potential nutraceutical fruit crop.

There was a broad variability in the data obtained for the breeding population; however, there were some orange pulp segregants, such as 84 (combined ZScore) and those from group 2 (pulp color-based-criteria), that can be considered as promising individuals for further breeding programs because of the preference with Ecuadorian consumers. In addition, there were other purple red pulp segregants that can also be considered as potential individuals to develop new varieties for exportation.

These results contribute to further tree tomato breeding programs focused on fruit quality traits.

## Figures and Tables

**Figure 1 plants-11-00268-f001:**
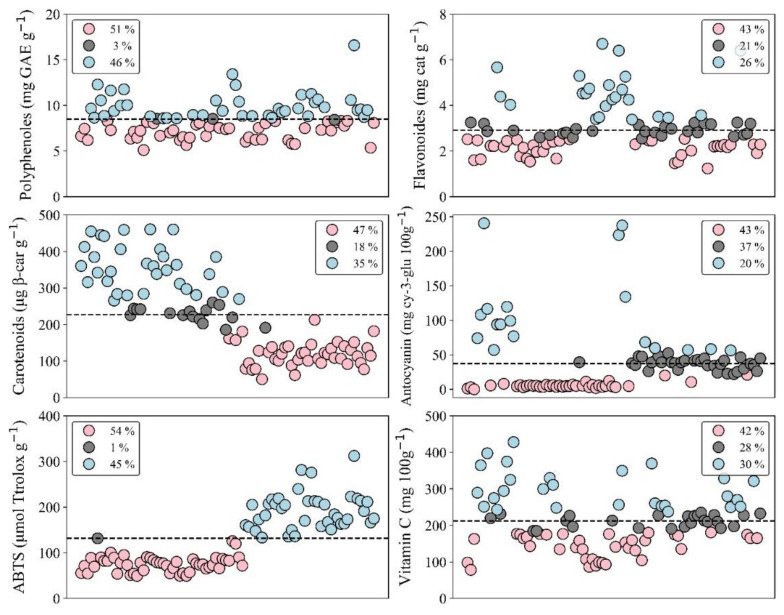
Data variability observed in some relevant traits from the tree tomato breeding population. Red color represents the low category, black the medium category and blue the high category (Table 2). The percentage of individuals belonging to each category is shown in the legend of each trait. The dotted line indicates the mean.

**Figure 2 plants-11-00268-f002:**
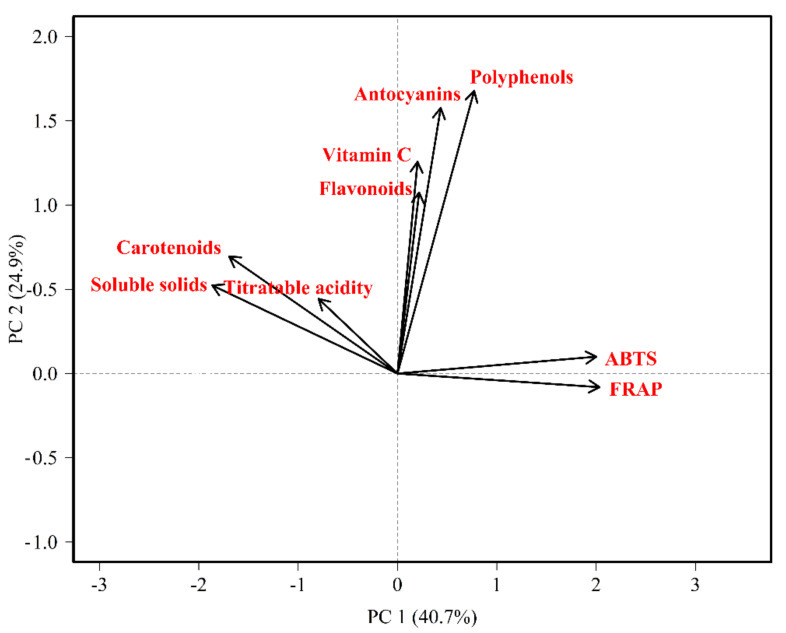
Principal component analysis for chemical fruit traits in a tree tomato breeding population.

**Figure 3 plants-11-00268-f003:**
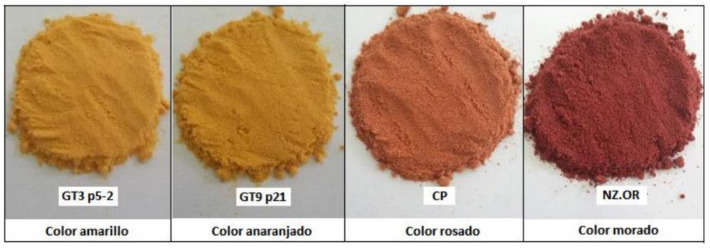
Different pulp fruit colors obtained from tree tomato samples of the breeding population.

**Table 1 plants-11-00268-t001:** Color and chemical parameters from the tree tomato breeding population.

Statistical Parameter	*L*	*a*	*b*	TA	SS	SAR	TI	TP *	TF *	TC *	TAC *	VC	FRAP *	ABTS *
Average	65.54	9.52	41.20	1.13	10.61	9.75	1.62	8.47	2.99	223.38	38.08	209.80	167.68	128.06
Minimum	33.54	3.12	3.65	0.68	7.00	6.07	1.20	5.11	1.24	50.39	1.06	78.29	52.43	49.51
Maximum	77.35	19.11	55.99	1.63	12.90	16.97	2.02	16.59	6.70	460.72	240.49	428.16	361.70	312.30
Variation (%)	56.64	83.67	93.48	58.28	45.74	64.23	68.3	69.20	81.49	89.06	99.56	81.71	85.50	84.15

*L* = brightness; *a* = coordinates red/green; *b* = coordinates yellow/blue; TA = titratable acidity (%); SS = soluble solids (°Brix); SAR = sugar/acidity ratio; TI = taste index; TP = total polyphenols (mg GAE g^−1^); TF = total flavonoids (mg cat g^−1^); TC = total carotenoids (µg β-carotene g^−1^); TAC = total anthocyanin content (mg cy-3-glu 100 g^−1^); VC = vitamin C (mg 100 g^−1^); FRAP = antioxidant capacity FRAP method (µm Trolox g^−1^); ABTS = antioxidant capacity ABTS method (µm Trolox g^−1^). * Parameters are given per dry weight.

**Table 2 plants-11-00268-t002:** Categories for the different chemical traits on the tree tomato breeding population.

	Categories		
Chemical Parameter	Low	Medium	High
Polyphenols * (mg GAE g^−1^)	5.11–8.39	8.40–8.54	8.55–16.59
Flavonoids * (mg cat g^−1^)	1.24–2.59	2.60–3.36	3.37–6.70
Carotenoids * (µg β-carotene g^−1^)	50.30–183.38	183.39–263.35	263.36–460.76
Anthocyanins * (mg cy-3-glu 100 g^−1^)	1.06–21.78	21.79–54.36	54.37–240.49
Vitamin C (mg 100 g^−1^)	78.29–183.91	183.92–235.67	235.68–428.16
FRAP * (µm Trolox g^−1^)	52.43–167.59	167.60–167.74	167.75–361.70
ABTS * (µm Trolox g^−1^)	49.51–127.97	127.98–128.12	128.13–312.30
Soluble solids (°Brix)	7.00–10.52	10.53–10.67	10.68–12.9
Trititable acidity (%)	0.68–1.05	1.05–1.12	1.20–1.63

* Parameters are given per dry weight.

**Table 3 plants-11-00268-t003:** Factors identified based on the variables evaluated in the tree tomato breeding population.

	Factors		
Variable	Antioxidant Capacity	Appearance	Maturity
ABTS	0.98		
FRAP	0.94		
TC	−0.80		
SS	−0.91		
*L*		0.94	
*b*		0.92	
TAC		−0.93	
TA			0.89
SAR			−0.96

TC = total carotenoids; TAC = total anthocyanin content; FRAP = antioxidant capacity FRAP method; ABTS = antioxidant capacity ABTS method; *L* = brigthness; *b* = coordinates yellow/blue; SS = soluble solids; TA = titratable acidity; SAR = sugar/acidity ratio.

**Table 4 plants-11-00268-t004:** Z Score values for the fruit chemical traits in the tree tomato breeding population. Bold indicates values ≥ 2.

Segregant	TP	TF	TC	TAC	FRAP	ABTS	VC	SS	TA	SAR	TI
4	0.62	−0.45	**2.00**	0.76	−0.92	−0.60	1.06	0.87	0.56	−0.22	0.70
5	0.10	−1.18	1.39	1.47	−1.04	−0.90	**2.05**	0.51	**2.15**	−1.25	**2.24**
6	**2.00**	0.19	1.02	**4.26**	−0.53	0.06	0.55	0.94	−0.08	0.40	0.21
7	1.09	−0.10	1.90	1.65	−0.86	−0.59	**2.49**	1.02	1.55	−0.80	1.69
10	1.66	**2.35**	1.04	1.17	−0.26	−0.43	0.44	0.94	1.21	−0.58	1.31
13	0.79	−0.48	1.57	1.71	−0.78	−0.77	**2.19**	1.46	**2.15**	−0.94	**2.48**
14	1.73	0.91	**2.02**	1.29	−0.58	−0.51	1.52	0.58	0.43	−0.22	0.45
15	0.82	−0.08	0.48	0.82	−0.73	−0.84	**2.89**	0.58	**2.02**	−1.20	**2.12**
22	0.17	−0.90	**2.04**	−0.69	−0.82	−0.60	−0.33	1.68	−1.59	**3.25**	−0.04
26	0.06	−0.25	1.40	−0.71	−0.80	−0.79	1.59	0.94	**2.15**	−1.11	**2.36**
29	−0.63	−0.47	**2.03**	−0.69	−1.01	−0.94	−0.99	1.02	−0.55	0.98	−0.10
35	0.26	**2.02**	−0.01	0.03	−0.74	−0.65	−0.68	0.65	0.48	−0.22	0.51
36	−0.31	1.35	0.49	−0.69	−0.79	−0.79	−0.99	0.65	−1.93	**3.17**	−0.53
37	−0.20	1.35	−0.07	−0.57	−0.83	−0.84	−1.36	0.94	−1.37	**2.14**	−0.47
42	1.08	**3.25**	1.39	−0.73	−0.60	−0.60	−1.49	0.80	−0.21	0.49	0.02
47	**2.60**	**2.99**	−0.03	**3.91**	0.02	−0.04	0.62	0.94	−0.68	1.07	−0.22
48	1.97	1.49	−0.57	**4.19**	−0.11	−0.12	1.85	1.02	−0.12	0.49	0.21
49	1.02	**1.99**	0.40	**2.02**	−0.32	−0.58	−0.75	0.72	−0.85	1.20	−0.41
57	−0.22	−0.46	−0.28	0.02	0.95	0.83	**2.12**	−1.54	−0.55	−0.58	−1.15
68	1.41	−0.10	−0.87	0.40	1.61	**2.36**	0.20	−0.59	−1.67	1.52	−1.33
71	1.46	−0.16	−0.67	0.10	1.03	**2.27**	0.24	−0.66	−1.20	0.76	−1.27
84	**4.27**	**2.99**	−0.62	0.18	**2.16**	**2.84**	0.55	−2.64	−1.80	0.00	**−2.57**

TP = total polyphenols; TF = total flavonoids; TC = total carotenoids; TAC = total anthocyanin content; FRAP = antioxidant capacity FRAP method; ABTS = antioxidant capacity ABTS method; VC = vitamin C; SS = soluble solids; TA = titratable acidity; SAR = sugar/acidity ratio; TI = taste index.

**Table 5 plants-11-00268-t005:** Z Score values, and color parameters of selected elite individuals.

Segregant	Z Sscore TP ∗ TC ∗ VC	*a*	*b*	Pulp Color	°*H*	*C*	TI
4	1.12	12.05	32.19	Orange	−1.97	34.37	1.73
6	1.42	12.63	7.56	Purple red	1.47	14.72	1.65
7	1.62	17.90	21.01	Purple red	0.42	27.60	1.89
10	1.23	12.35	25.62	Purple red	−0.55	28.44	1.83
13	1.30	16.70	20.15	Purple red	0.38	26.17	2.02
14	1.78	9.52	22.52	Purple red	−1.02	24.45	1.69
15	1.14	14.31	26.69	Purple red	−0.30	30.28	1.96
47	1.41	13.64	4.19	Red	3.15	14.27	1.58
48	1.19	13.69	3.65	Red	3.66	14.17	1.65
84	2.06	7.77	44.92	Yellow	−1.82	45.59	1.20

TP = total polyphenols; TC = total carotenoids; VC = vitamin C; *a* = coordinates red/green; *b* = coordinates yellow/blue; °*H* = Hue angle; *C* = Chroma; TI = taste index.

**Table 6 plants-11-00268-t006:** Analysis of variance of the grouping criteria by antioxidant compounds.

Group	Segregant	TP *		TF *		TC *		TAC *		VC		TA		SS		SAR		TI		FRAP *		ABTS *	
1	53, 58, 59, 60, 61, 62, 63, 67, 68, 70, 71, 73, 75, 83, 84, 85, 86, 87	9.96	ab	3.21	abc	114.97	de	41.73	c	220.83	c	1.03	c	9.16	b	9.48	b	1.49	d	292.81	a	224.77	a
2	6, 47, 48	12.64	a	4.76	a	239.62	bc	233.79	a	285.85	bc	1.06	bc	11.93	a	11.27	ab	1.63	bcd	149.16	c	125.81	c
3	4, 5, 7, 13, 14, 15, 26	9.89	abc	2.72	abc	402.39	a	85.51	b	358.49	a	1.50	a	11.77	a	7.96	b	1.90	a	94.49	d	81.63	d
4	1, 2, 3, 8, 21, 22, 23, 25, 29, 30, 40, 42	7.91	cd	2.84	abc	389.35	a	3.71	e	148.25	d	1.03	c	11.77	a	11.84	a	1.62	cd	84.93	d	75.88	d
5	51, 52, 54, 55, 56, 64, 65, 66, 69, 74, 76, 77, 78, 82, 89	6.85	d	2.19	c	100.44	e	33.86	cd	175.20	d	1.09	bc	9.30	b	8.93	b	1.53	cd	230.98	b	157.59	bc
6	57, 72, 79, 80, 81, 88, 90	8.60	bcd	2.48	bc	160.19	cd	39.96	cd	288.22	b	1.10	bc	9.38	b	8.63	b	1.54	cd	245.08	b	183.64	b
7	16, 17, 18, 19, 20, 32, 34, 35, 36, 37, 38, 39, 41, 43, 44, 45, 46, 49, 50	7.65	d	3.56	ab	234.67	c	14.02	de	147.54	d	1.11	bc	11.51	a	10.78	ab	1.64	bc	92.11	d	70.48	d
8	9, 10, 11, 12, 24, 27, 28, 31, 33	8.07	cd	2.88	abc	304.44	b	30.83	cde	256.95	bc	1.34	ab	11.83	a	8.93	b	1.78	ab	84.65	d	70.19	d

Different letters indicate significant differences between groups (LSD test, *p* ≤ 0.05). TP = total polyphenols (mg GAE g^−1^); TF = total flavonoids (mg cat g^−1^); TC = total carotenoids (µg β-carotene g^−1^); TAC = total anthocyanin content (mg cy-3-glu 100 g^−1^); VC = vitamin C (mg 100 g^−1^); TA = titratable acidity (%); SS = soluble solids (°Brix); SAR = sugar/acidity ratio; TI = taste index; FRAP = antioxidant capacity FRAP method (µm Trolox g^−1^); ABTS = antioxidant capacity ABTS method (µm Trolox g^−1^). * Parameters are given per dry weight.

**Table 7 plants-11-00268-t007:** Analysis of variance of the grouping criteria by pulp color.

Group	Segregant	TP *		TF *		TC *		TAC *		VC		TA		SS		SAR		TI		FRAP *		ABTS *	
1	33, 37, 42, 54, 55, 56, 63, 65, 69, 70, 73, 84	8.59	bc	3.42	ab	155.10	bc	31.11	cd	175.44	bc	0.98	b	9.66	ab	10.17	ab	1.49	b	219.24	ab	160.72	ab
2	35, 51, 66, 71, 72, 74, 78, 87, 90	8.32	bc	2.87	ab	131.54	c	.39.16	c	190.71	bc	1.08	b	9.26	b	8.83	ab	1.52	b	233.92	ab	175.59	ab
3	1, 2, 3, 8, 12, 16, 17, 18, 19, 21, 22, 23, 24, 25, 26, 27, 28, 29, 30, 32, 36, 40, 41, 52, 64, 67, 68, 75, 76, 77, 79, 80, 83, 88, T89	7.87	c	2.45	b	259.14	ab	15.37	de	200.64	bc	1.10	b	11.05	a	10.63	a	1.63	ab	142.52	b	110.72	ab
4	53, 58, 59, 62	8.91	abc	3.23	ab	114.67	c	52.04	c	229.79	abc	1.29	ab	9.10	b	7.10	ab	1.64	ab	295.59	a	206.83	a
5	6, 47, 48	12.64	a	4.76	a	239.62	abc	233.79	a	285.85	ab	1.06	b	11.93	a	11.27	a	1.63	ab	149.16	ab	125.81	ab
6	4, 9, 15, 57, 6061	8.99	abc	2.63	ab	241.77	abc	50.99	c	311.57	a	1.26	ab	10.45	ab	8.40	ab	1.68	ab	178.75	ab	142.20	ab
7	20, 31, 34, 38, 39, 43, 44, 45, 46, 50, 82, 85, 86	7.80	c	3.45	ab	203.35	abc	9.74	e	157.84	c	1.14	ab	10.88	a	9.68	ab	1.62	ab	127.68	b	100.52	b
8	5, 7, 10, 11, 13, 14, 49, 81	10.02	ab	3.57	ab	339.66	a	102.67	b	292.49	a	1.37	a	11.58	a	8.76	ab	1.80	a	125.54	b	97.03	b

Different letters indicate significant differences between groups (LSD test, *p* ≤ 0.05). TP = total polyphenols (mg GAE g^−1^); TF = total flavonoids (mg cat g^−1^); TC = total carotenoids (µg β-carotene g^−1^); TAC = total anthocyanin content (mg cy-3-glu 100 g^−1^); VC = vitamin C (mg 100 g^−1^); TA = titratable acidity (%); SS = soluble solids (°Brix); SAR = sugar/acidity ratio; TI = taste index; FRAP = antioxidant capacity FRAP method (µm Trolox g^−1^); ABTS = antioxidant capacity ABTS method (µm Trolox g^−1^). * Parameters are given per dry weight.

## Data Availability

Data are contained within the article.

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
