# Peer review of "Phytochemical Characterization of a Tree Tomato (*Solanum betaceum* Cav.) Breeding Population Grown in the Inter-Andean Valley of Ecuador"

_plants, 2022, doi:10.3390/plants11030268_

Round 1
Reviewer 1 Report
Major comments:
Abstract should contain at least partly concreate values of the parameters tested, at least concentration ranges
Materials and methods
The description of the methods used is poor.
1)Do the authors use the whole fruits or just a pulp? According to literature data peel is not edible.
2)Lines 95-96 ‘An amount equivalent to 200 g of each cultivar was homogenized and a volume of 30 mL of each sample was placed in a Petri dish’. How is it possible: first ‘g’ and then ‘ml’??? Besides it is not clear how the authors obtained the carotenoids concentrations (Table 1). Furthermore, are there any data on the carotenoid composition of tree tomato fruits??? Taking into account the color of lyophilized samples there should be a mixture of various carotenoids. Any literature data are desirable. I can’t understand how carotenoid content was determined.
3) Line 101- ‘VC (citric acid) was measured with a reflectometer RQflex plus”. First of all vitamin C is ascorbic acid, but not citric one. Then no description of ascorbic acid determination is provided. It is difficult to understand what the authors meant saying that they used reflectometer in this case. What does it mean???
4) determination of polyphenols and flavonoids As far as I know acidic extracts are never used in the determination of these substances, please explain the necessity of formic acid utilization in the extraction process
5)No data of the dry weight content are presented that prevents to understand the real nutritional values of genotypes.
6)According to ‘Plants’ rules ‘Material and Methods’ section should be put after ‘Results and Discussion’
7)What are the units of ‘cromaticity’?
Results and discussion
- There is no information about evaluation of fruit taste neither organoleptic nor via chemical testing. For characteristic of fruit quality the authors used Brix/TA values. This parameter is known to reflect the stage of fruit maturity for tomatoes. On the other hand, it seems rather attractive to try and use another parameter, the so called ‘Taste index’ (TI) widely used for taste evaluation of tomatoes (TI= TA+Brix/20TA) (Navez et al, 1999 Le criteres de qualite de la tomate. Infos-Clifl, Vol.155; P.41-47). Earlier we have used this parameter successfully for evaluation of physalis taste. It will be highly interesting to calculate TI values and make a comparison between the genotypes,
- As the concrete biochemical characteristics are produced by the authors only on Figure 2 it is desirable to add full Tables of the data in Supplement
- Peel/pulp antioxidants distribution is highly valuable for characteristic of plants. Are there any data on the topic?
- Please check points and commas in Tables and in the text. I fear that Table 1,3,5,6 should contain values with points but not commas (the same on line 242:
‘4270,00 µg β-carotene g-1’?????)
- It seems interesting to indicate percentage of samples without antocyanin content. Is it possible to calculate % of low, medium and high levels of parameters indicated on Fig.2?
- Reference list should contain ‘doi’ data
Minor comments:
- Line 357 ‘; SS= Sólidos solubles’- use English version
- Line 367 ‘. [36] reported an antioxidant capacity..’- don’t begin the sentence with the number of the reference
- Table 1.- please add everywhere that all the parameters are given per dry weight
- Line 131 ‘(mg GAE g-1 DW)’- change to ‘(mg GAE g-1 DW)’ (please check all the text)
- Line 127 ‘Na2CO3’ change to ‘Na2CO3’ (please check all the text; for instance, CH3COOH, FeCl3 etc)
Author Response
Reviewer 1
Comment: Abstract should contain at least partly concreate values of the parameters tested, at least concentration ranges
Reply: Data about results were added in the abstract.
Comment: Do the authors use the whole fruits or just a pulp? According to literature data peel is not edible.
Reply: It was clarified that pulp (including mesocarp, mucilage, seeds and placenta) was ground and sieved to obtain just the liquid pulp for the analyses because it is the edible part of the fruit.
Comment: Lines 95-96 ‘An amount equivalent to 200 g of each cultivar was homogenized and a volume of 30 mL of each sample was placed in a Petri dish’. How is it possible: first ‘g’ and then ‘ml’???
Reply: There was a mistake, it was corrected in the text.
Comment: Besides it is not clear how the authors obtained the carotenoids concentrations (Table 1).
Reply: The methodology for carotenoid determination has been added to the section of “Materials and Methods”.
Comment: Furthermore, are there any data on the carotenoid composition of tree tomato fruits??? Taking into account the color of lyophilized samples there should be a mixture of various carotenoids. Any literature data are desirable. I can’t understand how carotenoid content was determined.
Reply: In the “Results and Discussion” section some data in mentioned about carotenoid content in tree tomato based on different authors such as Acosta-Quezada et al. (2015), Rojas Benitez et al. (2017), Mertz et al. (2009) and Kou et al. (2009). In addition, some information about types of carotenoids found in tree tomato was also included (Diep et al. (2020) carotenoid paper).
Comment: Line 101- ‘VC (citric acid) was measured with a reflectometer RQflex plus”. First of all vitamin C is ascorbic acid, but not citric one. Then no description of ascorbic acid determination is provided. It is difficult to understand what the authors meant saying that they used reflectometer in this case. What does it mean???
Reply: there was a mistake and it has been corrected that vitamin C is related to ascorbic acid. The description about how was it was determined has been added.
Comment: Determination of polyphenols and flavonoids as far as I know acidic extracts are never used in the determination of these substances, please explain the necessity of formic acid utilization in the extraction process
Reply: Polyphenols are normally extracted in alcoholic solutions (methanol or ethanol) in water, acetone/water and with the addition of acetic acid or formic acid that help to improve the recovery. Therefore, in this research, a phenolic compound extraction process was optimized using 70/30 / 0.1 v / v / v methanol/water/formic acid solutions, establishing the number of extraction cycles needed to obtain 100% recovery; demonstrating that the solvent system used works correctly for the tree tomato matrix.
Comment: No data of the dry weight content are presented that prevents to understand the real nutritional values of genotypes.
Reply: The ranges of the content for each parameter are shown in Table 1 and all data has been added as supplementary table.
Comment: According to ‘Plants’ rules ‘Material and Methods’ section should be put after ‘Results and Discussion’
Reply: The section of “Materials and Methods” has been placed after “Results and Discussion”
Comment: What are the units of ‘cromaticity’?
Reply: In Materials and Methods, it was mentioned that chromatic properties were determined by the L* a* b* color method of the CIE and were expressed in terms of the L, a and b coordinates. In addition, the formulas to calculate Hue° and Chroma has been included. Chroma is calculated based on the color parameters a and b thus it has not units.
Comment: There is no information about evaluation of fruit taste neither organoleptic nor via chemical testing. For characteristic of fruit quality the authors used Brix/TA values. This parameter is known to reflect the stage of fruit maturity for tomatoes. On the other hand, it seems rather attractive to try and use another parameter, the so called ‘Taste index’ (TI) widely used for taste evaluation of tomatoes (TI= TA+Brix/20TA) (Navez et al, 1999 Le criteres de qualite de la tomate. Infos-Clifl, Vol.155; P.41-47). Earlier we have used this parameter successfully for evaluation of physalis taste. It will be highly interesting to calculate TI values and make a comparison between the genotypes.
Reply: Taste index has been calculated, all data is in the supplementary file. The range of this parameter has been included in Table 1 and data for selected segregants has been included in Table 5. ANOVA results has been also included. It has been included in the selected genotypes.
Comment: As the concrete biochemical characteristics are produced by the authors only on Figure 2 it is desirable to add full Tables of the data in Supplement
Reply: A table with all the data has been added as supplementary file.
Comment: Peel/pulp antioxidants distribution is highly valuable for characteristic of plants. Are there any data on the topic?
Reply: In this study just the pulp was analyzed but data about antioxidants composition of tree tomato peel was added in the discussion.
Comment: Please check points and commas in Tables and in the text. I fear that Table 1,3,5,6 should contain values with points but not commas (the same on line 242: ‘4270,00 µg β-carotene g-1’?????)
Reply: Dots and commas have been checked, commas has been replaced by dots. Data in line 242 was corrected.
Comment: It seems interesting to indicate percentage of samples without antocyanin content. Is it possible to calculate % of low, medium and high levels of parameters indicated on Fig.2?
Reply: the percentage of individuals belonging to each category for the different compounds has been added in Figure 2.
Comment: Reference list should contain ‘doi’ data
Reply: The instruction for authors of the journal not mention to add the doi for the references, that is the reason why we did not add doi.
Comment: Line 357 ‘; SS= Sólidos solubles’- use English version
Reply: The words have been written in english
Comment: Line 367 ‘. [36] reported an antioxidant capacity..’- don’t begin the sentence with the number of the reference
Reply: The sentence has been re-written and also in other parts of the text where the sentences started with the number of reference.
Comment: Table 1.- please add everywhere that all the parameters are given per dry weight
Reply: It has been added “Parameters are given per dry weight” in all tables were theses parameters are shown.
Comment: Line 131 ‘(mg GAE g-1 DW)’- change to ‘(mg GAE g-1 DW)’ (please check all the text)
Reply: It has been changed in all the document.
Comment: Line 127 ‘Na2CO3’ change to ‘Na2CO3’ (please check all the text; for instance, CH3COOH, FeCl3 etc)
Reply: Changes has been done in all the text.
Reviewer 2 Report
All my concerns about this manuscript are listed into the attached file plants-1483411_R1.

Author Response
Reviewer 2
Comment: Add reference of Corbe et al. (2019)
Reply: Reference has been added and included in the text of introduction.
Comment: Determination of vitamin C (VC), soluble solids (SS), titratable acidity (TA), and sugar/acid ratio (SAR), it is a subtitle?
Reply: It is a subtitle and it has been numbered.
Comment: Which is the relationship? specify
Reply: It has been specified how was calculated the relationship between SS and TA (sugar/acidity ratio) and the formula has been included.
Comment: Which were the US conditions (power, frequency, temperature, pulse, etc).
Reply: The characteristics of the ultrasonic bath were added.
Comment: But in line 120 you wrote the extraction process was repeated three folds. May you explain this discrepancy?
Reply: It was clarified that they were 5 cycles of extraction.
Comment: I do not understand your analysis, firstly you performed a multivariate clusterization by PCA then you carried out an univariate clusterization. Why?
Reply: It was clarified that PCA was carried out to visualize the relationship between antioxidants compounds and soluble solids and tritatable acidity, taking into consideration the correlation among variables. However, to carry out the univariate analysis (ANOVA), the k-means method (which takes also into consideration the Elbow method) was used to form specific grouping based just in antioxidants compounds (first grouping) and color parameters (second grouping). The antioxidant compounds used were polyphenols, flavonoids, carotenoids, anthocyanins, FRAP and ABTS and vitamin C while the color parameters were L, a and b. This explanation has been clarified in the “statistical analysis”. We did not use PCA for grouping antioxidant compounds and color separately because the PCA decrease the explicative power as much the number of parameters decrease, consequently the k-means method is more accurate.
Comment: How were the percentages assigned?
Reply: These are relevant desired traits according to the literature. This has been explained (literature added) in the corresponding section in the manuscript.
Comment: Rephrase more clearly lines 223 to 225.
Reply: These lines were re-written.
Comment: Was 6.70 lower than 6.44? Maybe, you mean the average value, so I think you must rephrase.
Reply: It was corrected because the comparison was with the average value.
Comment: Are you sure it is the right reference? This work deals with phenolic and anthocyanin compounds.
Reply: The data was reviewed and corrected and the right reference was Vasco et al. (2009).
Comment: Please check ref. [36] and [39]. Are they appropriate?
Reply: References were checked and they were corrected because the right references were Mutalabid et al. (2017) and De Rosso et al. (2007); in addition, the data was corrected.
Comment: “was a contrast between carotenoids and soluble solids vs antioxidant capacity” what does it measn?
Reply: The loading values have been added to show that carotenoids and soluble solids are opposite to antioxidant capacity. It means more antioxidant capacity less soluble solids and carotenoids in the pulp.
Comment: This statement is not supported by Fig3.
Reply: Figure 3 has been calculated again and was modified. It has been clarified that polyphenols are the main compounds related to antioxidant capacity and it supported by the value of the partial correlation among these two parameters. This has been added in the text. A table with the partial correlation values is added as supplementary table.
Comment: Can it depend on the nature of antioxidant assay? I mean that FRAP and ABTS should be assays for hydrophylic compounds. Anyway, from the vectors disposition, polyphenols do not influence antioxidant activity, while carotenoids are inversely related. What is your explanation?
Reply: PCA was done again and the plot changed. It has been clarified that partial correlation value support that polyphenols (0.52) is the main compound related to antioxidant capacity (ABTS), with carotenoids the correlation was not statistically significant and was too low (-0.11). This was clarified in the text. A table with the partial correlation values was added as supplementary table.
Comment: Please, better describe PCA results. For instance, what about the clusters?
Reply: PCA was done again because Vitamin C was not included and also there was a mistake in the last calculation (figure that you reviewed). The percentage of explanation for the components has been added, as well as the loading value for the vectors influencing the PC1. In addition, the partial correlation value that support that polyphenol is influencing the antioxidant capacity. The clusters were not explained because for clustering the k-means method was used for a specific grouping purpose (antioxidant and color criteria) and the ANOVA was used to analyses the groups.
Comment: Change PCA with PC.
Reply: This has been changed in Figure 3.
Comment: How come do not polyphenols influence antioxidant activity?
Reply: Polyphenols do influence in the antioxidant capacity and it is supported by the correlation value (0.52). In the factorial analysis, the interpretation of the results has been re-written and it can only be observed that carotenoids are not influencing the antioxidant capacity.
Comment: I wonder why you chose this criterion if TP, TC and VC do not influence antioxidant activity, as shown above?
Reply: It was explained the reason why these parameters were chosen based on the literature. In addition, TP was the only compound that influenced antioxidant capacity.
Comment: what do you mean line 364
Reply: It was clarified that segregant 84 got the highest Z score value (p<0.01) and it is a superior individual, especially because of its great amount of polyphenols.
Comment: Re-phrase lines 413 – 415
Reply: the sentence was re-written.
Comment: Which is the difference between Table 7 and Figure 2? Maybe, it would more appropriate to group this section to lines 217 - 271.
Reply: This section was grouped according to the suggestion.
Comment: Text changes made in the pdf file.
Reply: All changes have been done in the text.
Reviewer 3 Report
The topic of this research is particularly relevant in the field of safe and healthy food. The manuscript is important in practical terms also.
The title corresponds to the accomplished investigations.
Abstract presents the results of this study clearly and concisely.
Keywords are appropriate.
Introduction. Different sources were described, the aim of investigations was formulated however the state-of-the-art of investigations was not covered suitable.
Materials and methods. In the Introduction, the authors mentioned that the Fruit Program has been working on S. betaceum x S. unilobum hybrids. However, the Plant material (2.1. section) does not indicate which genotypes were selected for these studies.
Please explain if VC is vitamin C (p.3, line 99) why it was detected with a reflectometer as citric acid (line 101)?
Results are presented in seven tables and three figures. Different authors are very widely cited in the Results section (pp. 5-6), so the results are hardly analysed. This part of Results needs to be corrected. Otherwise, the data obtained by other authors seem to be compared only with each other. However, the results in sections 3.1, 3.2, 3.3, 3.4 and 3.5 are discussed more clearly and consistently.
Conclusions. The conclusions are exceptionally concise, but essentially in line with the stated aim.
Minor remarks.
- The figures should be clear to the reader without the main text of the manuscript. Fig. 2. is inadequate. What the different colours mean as they were not specified in section 3.5 either.
- Writing of species names. The names of the plant species must be written in Italic.
Author Response
Reviewer 3
Comment: Different sources were described, the aim of investigations was formulated however the state-of-the-art of investigations was not covered suitable.
Reply: The state-of-art has been mentioned in the introduction.
Comment: In the Introduction, the authors mentioned that the Fruit Program has been working on S. betaceum x S. unilobum hybrids. However, the Plant material (2.1. section) does not indicate which genotypes were selected for these studies.
Reply: The history of the breeding population has been added to the section of Plant Material.
Comment: Please explain if VC is vitamin C (p.3, line 99) why it was detected with a reflectometer as citric acid (line 101)?
Reply: This was a mistake, it was measured as ascorbic acid. This has been corrected in the manuscript.
Comment: Results are presented in seven tables and three figures. Different authors are very widely cited in the Results section (pp. 5-6), so the results are hardly analyzed. This part of Results needs to be corrected. Otherwise, the data obtained by other authors seem to be compared only with each other. However, the results in sections 3.1, 3.2, 3.3, 3.4 and 3.5 are discussed more clearly and consistently.
Reply: The results section has been improved.
Comment: The figures should be clear to the reader without the main text of the manuscript. Fig. 2. is inadequate. What the different colours mean as they were not specified in section 3.5 either.
Reply: The section 3.5 (compound categories) was join with the results related to the chemical parameters. Figure 2 was modified and clarifications has been placed in the figure´s title.
Comment: Writing of species names. The names of the plant species must be written in Italic.
Reply: The whole document has been checked and all scientific names have been written in Italic
Round 2
Reviewer 1 Report
Thank you for the revision. Just several minor comment:
Line 96) ‘The results of this study were higher than those (1.17 to 1.91 mg GAE g-1) found by [34]..’ change to ‘the TP values found in this study…’
Line 106 ‘On the other hand, flavonol is a type of flavonoid compound that has only be detected in the peel of tree tomato’ change to ‘On the other hand, flavonol is a type of flavonoid compound that has only been detected in the peel of tree tomato..’
Line 110 ‘have also found’ change to ‘were also found’
\line 195- delete one ‘and’
Author Response
Comment: Line 96) ‘The results of this study were higher than those (1.17 to 1.91 mg GAE g-1) found by [34]..’ change to ‘the TP values found in this study…’
Reply: Change was done.
Comment: Line 106 ‘On the other hand, flavonol is a type of flavonoid compound that has only be detected in the peel of tree tomato’ change to ‘On the other hand, flavonol is a type of flavonoid compound that has only been detected in the peel of tree tomato..’
Reply: Change was done.
Comment: Line 110 ‘have also found’ change to ‘were also found’
Reply: Change was done.
Comment: line 195- delete one ‘and’
Reply: It has been deleted.
Reviewer 2 Report
Just minor comments are grouped in the attached file plants-1483411_R2.

Author Response
Comment: Changes made in the pdf file.
Reply: All changes have been done in the text.
Comment: Which flavonol? Specify it
Reply: Quercetin and myricetin have been mentioned in the text.
Comment: Value for TI variation.
Reply: 68.3% has been added to the table 1.